# Professional Teacher Identity on the Boundary: Student Study Group Facilitators Negotiating Their Identity

**David R. Arendale** [1,*] and **Amanda R. Hane** [2]

1   Curriculum and Instruction Department, University of Minnesota-Twin Cities, Minneapolis, MN 55455, USA
2   Twin Cities Public Television, Minneapolis, MN 55455, USA; amandahane@gmail.com
*   Correspondence: arendale@umn.edu; Tel.: +1-612-812-0032

**Abstract:** This study focuses on undergraduate student paraprofessionals who facilitated peer study groups for academically challenging college courses. A grounded qualitative research study of these student facilitators at an institution identified their professional identity development in unexpected ways that went against written policies for teacher attitudes and behaviors. Rather than perceiving themselves as peer study group facilitators following a narrow job description, many of them referred to themselves as teachers and acted accordingly, breaking the boundaries of behavior established by the study group program administrator. This article unmasks this professional identity emergence, postulates the mechanism for this choice, presents a model to explain it, and makes recommendations for talking openly about this identity and the subsequent behaviors that occurred as a result by the facilitators and the implications for the PAL program.

**Keywords:** peer-assisted learning; college/tertiary education; teacher identity emergence; personal and professional development; student paraprofessionals; learning assistance; academic study groups; professional identity development

## 1. Introduction

The undergraduate postsecondary experience is a powerful and rich environment for student development academically, socially, and personally. This takes place in the classroom, in co-curricular activities, in work activities, in interpersonal relationships, and through personal reflection. This study focuses on a structured peer study group program involving students enrolled in historically difficult academic courses. For context, the following introduction provides an overview of the Peer Assisted Learning (PAL) program, identifies the specifics of the PAL program, reviews the detailed training program for the PAL student paraprofessionals who provide direct facilitation of the study sessions, and highlights the prohibitions against PAL student study group facilitators behaving as a teacher during the study sessions.

### 1.1. Overview of the Peer Assisted Learning (PAL) Program

The host university for the PAL program is a public research-intensive institution with 50,000 students enrolled in undergraduate, graduate, and professional school degrees in the United States. Despite selective admissions standards, administrators were concerned about unacceptable dropout rates. During the fall of 2006, the Peer Assisted Learning (PAL) program was created to increase student graduation rates and persistence in academically rigorous programs such as science, technology, engineering, and mathematics (STEM) [1]. Regardless of graduation rate improvement, students switching from degree programs in STEM to other, less rigorous ones undermine institutional efforts to increase the diversity and success of historically underrepresented students [2]. PAL targets historically difficult math and science courses for first- and second-year students with high rates of final course grades of D, F, I, or withdrawal [1]. These academically challenging courses are gatekeepers

before students are permitted to enter highly competitive academic programs in STEM and other professional programs at the undergraduate, graduate, and professional school levels. Following is a description of the PAL program that is similar to others previously published by the authors of this article [3].

PAL incorporates the best practices of Supplemental Instruction-PASS (SI-PASS) [4], Peer-led Team Learning (PLTL) [5], and the Emerging Scholars Program (ESP) [6]. Participation in PAL was mandatory for students in the targeted gatekeeper classes. They were required to participate in weekly study review sessions, and their attendance was reported to the target course's professors and the campus PAL administrator to track compliance. Irregular attendance often had a negative impact on the final course grade. Several studies have validated the efficacy of the PAL program in increasing student academic success [3]. PAL attendance was uniformly high and regular throughout the academic term. These sessions were led by an upper-division undergraduate student employed by the campus learning center. Due to the mandatory attendance policy, the PAL sessions ranged from 20 to 30 student participants.

*1.2. The PAL Facilitator*

The student leaders were competitively selected based on academic competency in the content area, previous success in the targeted gatekeeper courses for which they would serve as study group facilitators, and good interpersonal communication skills. Twenty percent of the facilitators were concurrently enrolled in an education major or planned to do so. The position attracted their attention due to the salary and the belief that it served as an early teaching experience. Most of the remaining 80% of PAL facilitators planned to complete doctoral degrees in graduate or professional schools, a common aspiration among many admitted students. Their self-reported interest in the PAL job included (a) reinforcing basic subject area concepts needed for later courses in their major, (b) genuine interest in helping other students to be successful in the introductory course, (c) the salary, and (d) belief that the job experience would be helpful for admission to graduate or professional schools in programs such as law, medicine, STEM, and other health and science degrees. Most thought that the PAL job experience was viewed positively when committees evaluated admission applications as evidence of their communication and small group skills. For students planning to attend graduate school, most thought the PAL experience provided training to serve as a graduate teaching assistant (GTA), which is often required to obtain financial aid for a doctoral degree. Students believed teaching as a GTA would be a short-term, part-time job during graduate school to support their long-term objective as commercial or university researchers.

To clarify the student leader's role during PAL sessions, their formal title was "facilitator" rather than "leader". During the academic term, an important objective for PAL sessions was for the facilitator to transfer leadership in the review sessions from themselves to study group participants. One reason for this was the intentional use of varied collaborative learning activities during PAL sessions so all students could become leaders and actively engage with the group's work.

In addition to a review of course content during PAL sessions, the PAL facilitator modeled two types of learning strategies. Cognitive strategies included lecture note-taking, effective reading, visual organizers, and exam preparation. Due to the requirements of the math and science courses, extensive time was spent on problem-solving strategies using preplanned worksheets. These contained different types of problem sets to complete individually and collaboratively during PAL sessions with the assistance of the PAL facilitator. Metacognitive strategies included identifying error patterns in exams, selecting appropriate cognitive strategies based on the learning task, and self-testing for comprehension. Students applied these skills to course material during study sessions so they became a part of their academic repertoire when dealing with this and future courses. PAL facilitator training materials were available for review online [1,7].

*1.3. PAL Facilitator Training*

Facilitators engaged in extensive training before and during the academic term. A two-day workshop preceded the academic term, where they received basic instruction in PAL session procedures, participated in mock PAL sessions assuming the roles of facilitators and students, and worked in small groups to plan PAL sessions and reflect on choices made. Multiple times throughout their initial and subsequent training, the PAL facilitators were instructed to never assume the role of a teacher or teach a lesson. The reason for this explicit order was the political negotiation between the PAL program and the instructional staff, the result of which was that the PAL facilitators were never to assume the instructor's role. It was explained to the facilitators that on other college campuses, programs like PAL had been shut down due to complaints from the instructional staff that the student assistants were assuming the role of a classroom instructor, which was a violation of their role boundary.

The PAL training manual [7] was divided into eight principles that guided PAL facilitators during their study sessions. Principle #4 was that they were to shift PAL session authority and ownership over to the participants during the academic term as quickly as possible. During the fall term, they were required to enroll in a one-credit course (PsTL 3050) taught by a faculty member who co-developed the campus PAL program and was an advisor to the PAL program administrator. Students read assigned educational theory articles applicable to the PAL program and engaged in guided group discussion, discussed PAL participant behaviors, created solutions to improve the learning environment, and engaged in open-ended discussions of their growth due to the PAL experience.

As part of the PsTL 3050 course requirements for all new PAL facilitators, they maintained a weekly journal of their experiences and observed behaviors and perceptions of attitudes displayed by participants in their study groups. In addition, they also recorded reflections on their growth academically and personally. Facilitators noted changes in their students and themselves over the academic term and recorded examples in weekly journal entries. The PsTL 3050 course instructor read all entries and provided feedback and suggestions as warranted. Twice each academic term, they observed the PAL sessions of another facilitator, and then the two debriefed decisions made and reactions by participants. They had numerous opportunities for informal conversations with one another regarding what they had experienced and learned from their PAL experience. Twenty-five of the facilitators were interviewed regarding their PAL session experiences in the PsTL 3050 course as part of an assignment that included audio recordings to be used in a podcast. The course instructor for PsTL 3050 maintained notes on class discussions of the study group leaders concerning their attitudes and behaviors displayed during the study group sessions. A professional staff member administrated the PAL program, conducted periodic team meetings with the student staff throughout the academic term, periodically observed PAL sessions, coached and mentored the PAL student leaders, and performed program evaluations each term. PAL facilitators completed a detailed written reflection about their job experience and personal and professional growth at the end of the term [8].

As a grounded qualitative research study of the impact of the PAL experience on the student facilitators, the researchers did not have predispositions regarding what themes would emerge from the analysis of the data collected. Therefore, no formal research questions or hypotheses were identified before the qualitative data analysis. It was clear from previous research studies at other institutions that the PAL experience helped to foster personal and professional outcomes for both the participants and the facilitators. However, the area of identity formation by student study group leaders, or as they are called in this study, facilitators, was not extensively explored in the professional literature. The following section explores the previous research for possible guidance in analyzing the qualitative data derived from the open-ended survey questions.

## 2. Review of the Professional Literature

Since this was a grounded qualitative research study, the researchers had no preconceptions of the possible outcomes that would emerge from the student data. They were surprised by the emergence of a temporary professional identity formation as a teacher during the time that the students served as study group facilitators. Since this finding had not been reported extensively in previous research studies, the authors were required to conduct an extensive search of the professional literature for guidance regarding the possible mechanisms for the formation of this identity.

The professional literature regarding student study group facilitator identity development was conducted in seven phases to discover relevant scholarship and models for data interpretation: (a) college tutors and student study group leaders, (b) full-time professional college/tertiary tutors, (c) secondary school and college/tertiary teaching assistants, (d) secondary student teacher candidates, (e) novice and experienced secondary teachers, (f) communities of practice, and (g) classic theories of student identity development. Our first area to explore was student paraprofessionals in roles like the PAL facilitators.

### 2.1. College Tutors and Study Group Leaders

The first phase of the literature review of professional identity emergence examined college/tertiary student study group leaders and tutors like the PAL facilitators. Only four research studies reported on the development of tutoring or small group leader professional identities; however, the study group leaders or tutors did not perceive themselves as possessing an instructor identity [9–12]. Some of the studies found that the students began to assume the characteristics of the teaching staff. Brown et al. [10] studied undergraduate pre-service teachers who worked as writing center tutors. They had difficulty not crossing officially prescribed boundaries and acted as instructors rather than facilitators. "Yet, several facilitators found it difficult to 'draw the line' between effective facilitation and effective teaching. . ." (p. 12). The researchers recommended changes in tutor training so they could negotiate the ethical dilemma of helping students beyond their job description. As in the previous study, the tutor's future career aspiration of being a secondary school teacher appeared to complicate their interaction with students and not cross job role boundaries.

Clark and Raker [11] studied student facilitators in a Peer Led Team Learning (PLTL) program in an introductory organic chemistry course. When prompted to describe their role as a PLTL facilitator, some described themselves as teachers, which is officially prohibited by the program. Others described their roles as guides/facilitators and more as mentors. The student PLTL leaders explained how the environmental conditions of the PLTL sessions forced them into the role and expression of a teacher's professional identity. A frequent comment was that the PLTL participants were often unprepared to engage in academic conversations. "[I] often notice myself teaching the concept at the beginning of class because most of the students hadn't watched the video or had no clue about the material" [11] (p. 184). This theme of professional identity emergence as a response to the environmental conditions within the learning ecosystem rather than an initial predisposition toward the identity is repeated by other researchers. The result of many of these studies was that identity emergence occurs along a continuum. The interactions with students may pull a person toward an identity while the person pushes themselves toward it through repeated reflections throughout the academic term [13].

In a review of nearly 2000 publications related to postsecondary peer cooperative learning groups [3], only 4 studies identified the emergence of teaching staff professional identities. This database of 2000 publications only included research on multi-student peer study groups and not studies on individual tutoring. There were no similar studies on teacher identity formation within one-on-one tutoring programs. Due to the paucity of studies directly about student-led study groups, the researchers expanded their search for applicable studies to other education professionals to identify mechanisms that fostered the identity emergence of temporarily perceiving themselves as a teacher.

### 2.2. Full-Time Professional College/Tertiary Tutors

Several studies examined professional tutors during the second phase of this literature review. These full-time professional college/tertiary tutors were positioned between the course instructor and a student-led study group like the PAL program, which is the subject of this research study. These full-time tutors were typically found within British-style postsecondary or tertiary education. These tutors prepared students academically for the lectures delivered by the course instructor or professor. Compared to tenured professors, these tutors were considered marginal members of the academic community. They had multiple and sometimes conflicting identities. They switched identities as the situation demanded based on the needs of the students, thereby creating conflict between officially sanctioned work behaviors mandated by their supervisors and constructing a different identity to serve students. The mechanism for new identity emergence was tutor participation in a community of practice (CoP) [14,15] occupied by fellow tutors. These informal CoPs provided an environment for private conversations among the tutors regarding their work responsibilities and identity outside of observation by course instructional staff. However, CoPs have not been the subject of careful research concerning identity formation.

One British tutor in a qualitative study by Colley et al.s [16] stated she was "...in a fragmented group of teachers who forged '*a kind of plankton*', '*below the radar*'. Out of the gaze of management, the margins offered a space in which she enjoyed relative autonomy to teach according to her own professional principles..." (p. 9). The research illuminated the turmoil of the British tutors as they struggled to operate within the prescribed job duties assigned to them while developing their own professional identity. This led to stress due to "shutterings" [16] (p. 4) as they moved among multiple identities to please their superiors and meet the needs of their students. The tutors felt unresolved about successfully embracing either identity.

Research on British tutors by Ashcroft and Foresman–Peck [17] identified a common theme in the tutors: they were compelled to act on behalf of the best interests of the tutees, regardless of violating boundaries established by academic officials. Breaking the rules was seen as their highest moral decision. The researchers postulated that tutors needed to be evaluated regarding their moral choices as well as their tutoring behaviors compliant with national standards. This working environment forced the tutors to hide their true attitudes and behavior from their course instructors, professors, and administrators.

James and Diment [18] echoed that British tutors had "gone underground" and occupied a secret terrain not recognized or validated by the administration. One tutor expanded his/her job description to engage in "short episodes of 'teaching' and the provision of extra [academic content] materials..." (p. 414). The tutor justified this and other actions as fundamental for student success. The researchers described one tutor as a "humanistic practitioner" whose "...selflessness jeopardizes the time she has set aside for completing her own part-time higher degree" (p. 417). The researchers identified the lack of a match between the rhetoric of what the job required and the reality of tutors meeting student needs. Student needs "pulled" tutors into a teacher identity since it was the "humanistic" thing to do. Professional identity development occurred outside formal training or officially approved tutor job duties. Instead, it emerged through informal systems and conversations with fellow tutors that they called "...' underground' maintenance of practices that explicitly address learning" (p. 419). Another term for this communication among the tutors was an unofficial community of practice. The tutors took additional work underground since "...the practice in question may be somewhat at odds with orthodox or officially-sanctioned reifications" (p. 420). The researcher asked, "...would it be better for all concerned if the extensive learning activities were made visible, properly resourced, recognized?" (p. 420). To gain a broader perspective regarding teacher professional identity, the review of the professional literature expanded into three additional job categories. The first to be examined were secondary school and college/tertiary teaching assistants.

## 2.3. Secondary School and College/Tertiary Teaching Assistants

Phase three of this literature review examined teaching assistants (TAs) in secondary schools and postsecondary/tertiary institutions. One study [19] investigated the use of TAs in a Hong Kong secondary school. The TAs were professional staff assigned to assist the classroom teacher. These TAs expressed a disconnect between the way they talked about their teacher identity and how they manifested it in the classroom. It was not fixed, but rather, "…identity is a dynamic process" (p. 31). They felt that they were neither a teacher nor a teaching assistant but something in between. The researcher cited the solution proposed by Bhabha [20] to create a third space, which was not a compromise or hybrid but a new space and identity respected by all. This avoided rigid dichotomies and recognized multiple identities operating simultaneously within the class.

Graduate teaching assistants (GTAs) are doctoral candidates whose career goal is to work in private industry or higher education as researchers or college professors. A study by Dunn–Haley and Zanzucchi [21] argued that professional identity for GTAs formed as they negotiated the boundaries of simultaneous roles as a graduate student, a senior professor mentee, a novice researcher, and a part-time college instructor. These concurrent and often conflicting roles made it difficult for them to develop a single identity, especially since this process was not formally developed by their supervisors or formal coursework. Most GTAs are focused on developing their subject-area expertise and not on pedagogy or teacher identity development. Without formal pedagogy training, it is natural for most of them to replicate the teaching practices they experienced previously and concurrently as a student. Based upon their research with GTAs in the United States, researchers advocated for a carefully coordinated program of GTA orientation, continuing workshops, and careful mentorship by senior faculty members as key to developing their effective teaching persona. After gaining valuable insights from this first job category in secondary school and college/tertiary graduate assistants, the search for teacher identity formation was expanded to secondary school student-teacher candidates.

## 2.4. Secondary School Student-Teacher Candidates

Phase four of the literature review covered secondary school student-teacher candidates. A research study in The Netherlands [22] examined student teachers. The critical catalyst for identity emergence was not mentoring by the hosting classroom teacher or observing their behavior but by the community of practice formed with the other teacher candidates. The interaction of student teachers with each other and deep reflection about their experiences was fundamental for teacher identity emergence. However, the detailed process that operated within the CoPs was not explored. A study of pre-service teachers in Singapore by Chong et al. [23] found that teacher identity formation was mostly autobiographical and self-referential rather than externally influenced by teacher education or experiences during their student teaching experience.

Beauchamp and Thomas [24], in their study on student teachers in Canada, confirmed the importance of reflection on prior teaching activities and imagination of their future selves as catalysts for teacher identity formation. Their qualitative study affirmed previous findings by researchers about the influence of autobiography as a determinant of future vocational identity. Reflecting on the future was as important as prior behaviors and choices. The creation of their future identity drew student teachers toward its actuality. Sexton [25] positioned teacher education within a larger ecosystem of experiences and interactions for the student teachers. His research focused on the intersections of identity, role, and agency within this larger ecosystem. Sometimes, the emerging identity was at odds with formal teacher education, thereby creating a dissonance that led to students questioning their career choices. Antonek et al. [26] identified teaching portfolios as a comprehensive system for students to reflect upon practice and provide evidence of effectiveness. They also viewed them as "…portraits of teaching…" (p. 15) that became powerful tools for professional development. They described how the portfolio became a structured form of meditation for the students to enlarge the lens of introspective analysis and identity

confirmation. The portfolio reflection cycle needed the supervising teacher to respond to the student's words to deepen their contemplation of future reality. The final job category for investigation was secondary school teachers.

### 2.5. Novice and Experienced Secondary School Teachers

The fifth phase of the literature review explored the differences and similarities between novice and experienced secondary school teachers. Studies on teachers at the secondary level were reviewed. Each of the research studies was cited 300+ times in the professional literature and presented a comprehensive model for understanding teacher identity development. After a review of the debate about behavioral traits and character qualities of a good teacher, Korthagen [27] presented a theoretical model incorporating the importance of both through a hierarchical arrangement. His 'onion model' displayed important factors and arranged them in levels of observation by others. Professional identity development was initially shaped by the teacher's life experiences as a student and mentorship by other teachers. Failure to begin teacher education with an autobiography of the teacher candidate resulted in only temporary external compliance with expected behaviors. Congruence among inner mission, identity, and beliefs were necessary for lasting competencies and behaviors. "Professional identity, then, often takes on the form of a Gestalt: an unconscious body of needs, images, feelings, values, role models, previous experiences and behavioral tendencies, which together create a sense of identity" (p. 85). Despite considerable effort within teacher training programs, "...identity change is a difficult and sometimes painful process, and often there seems to be little change at all in how teachers view themselves" (p. 85). He argued that new teachers must choose their own identity rather than adopt one advocated by others.

Beijaard et al. [28,29] conducted a study on secondary teachers in The Netherlands regarding professional teacher identity development. Their model of identity contained three factors: the teacher as a subject matter expert, the teacher as a pedagogical expert, and the teacher as a didactical expert [Figure 1]. This model will be the most helpful for guiding analysis of the data from this qualitative research study since it traces the change in attitudes and behaviors on a continuum between novice and experienced secondary school teachers. Figure 1 displays the secondary teacher's development from the novice to the experienced level.

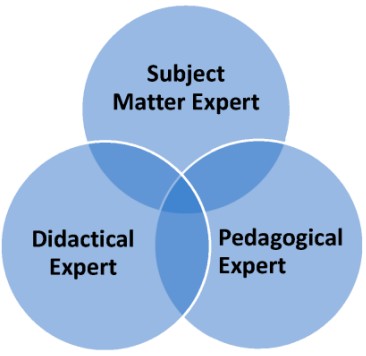

> Most Novice Teachers Begin with Exhibiting Subject Matter Expertise in their Behaviors with Students.
>
> Most Experienced Teachers Move to Exhibiting More Didactical and Pedagogical Expertise Behaviors with Students and Less on Displaying their Subject Matter Expertise.

**Figure 1.** Model of Professional Identity among Novice and Experienced Teachers.

In this model, subject matter expertise was defined as extensive knowledge of the academic subject material. This could be measured through subject-specific examinations. Novice secondary school teachers devoted most of their energy and action in the classroom to lectures that displayed their subject matter expertise. There was little interaction between novice teachers and the students and also a lack of engagement of the students with one another in the classroom. This behavior changed for experienced teachers. The experienced teachers instead focused more energy on employing pedagogical and didactical expertise to prepare students for a more engaged and productive learning environment. These

teachers sought to empower the students to develop their own subject-matter expertise, which could be demonstrated on the subject-specific examinations. Pedagogical expertise was viewed as the humanistic relationship the teacher establishes with students that takes into consideration their educational and cultural background. Establishing productive learning relationships is assumed to be a prerequisite for content learning to take place. Didactical expertise involved changing the learning dynamic for the teacher to facilitate engaged learning by students with each other rather than providing direct instruction to them passively. The next phase of the literature review briefly focused on Communities of Practice that can influence an individual's professional identity.

*2.6. Community of Practice*

Several previous research studies cited a Community of Practice (CoP) as a catalyst for the professional identity formation of tutors and study group leaders. A CoP is a group of people who work together on an ongoing activity or practice. Jean Lave and Etienne Wenger [14] developed the concept, and it was further refined by Wenger [15]. A CoP is an organic social theory of learning as opposed to formalized systems often promoted by educational institutions. A CoP can exist within formal institutions, but they are invisible to everyone except members of the CoP. Wenger [15] identified three elements of the groups, "...communities of practice, boundary processes among these communities, and identities as shaped by our participation in these systems" (p. 225). They were self-regulating and distributed power and leadership among group members. A CoP is a powerful ecosystem and mechanism for personal and professional development, as attested by the previously cited research. While the general principles for their operation have been identified, detailed studies on their inner workings have largely been ignored. They are difficult to assess and observe due to their invisibility to others who are not part of the CoP, which is populated by tutors and study group leaders. Our final phase of the literature review concludes with a brief overview of relevant theories on student identity formation.

*2.7. Classic Theories of Student Identity Development*

A wide range of theories explain student identity development. Due to space limitations, Erikson and Marcia were reviewed due to their direct relevance and helpfulness for the data interpretation of this study. Erik Erikson [30] identified eight stages of identity development. Stage Five is identity versus identity diffusion (confusion). Young adults seek support from those around them for identity formation. They experience confusion if older adults do not affirm them, and stress consequently occurs. This is especially applicable to student-led peer study group programs. Programs with consistent results are often infused with observation, mentoring, and discussions with full-time professional staff and instructional staff. These occasions provide opportunities for affirmation of identity formation by the student study group facilitators through their job activities.

James Marcia [31] developed a theory to more deeply explore Erikson's fifth stage. He stated that identity emerges from a crisis and the youth's commitment to resolve it. The young person considers several identities as alternatives to solve the crisis. During the crisis, the youth seeks trusted others to explore their options. Marcia stated that the young person selects from among four choices during a crisis: (a) Foreclosure: the person considers alternatives but abandons them and follows authorities without challenge; (b) Moratorium: while the person questions the authorities, no action is taken other than following the prescribed action; (c) Identity Achievement: the person considers alternatives and selects an identity other than the one endorsed by the authorities. While Marcia endorsed this choice as the healthiest state of mind, it also requires the highest risk-taking with authorities; (d) Diffusion: the person lacks commitment to much of anything and follows authorities and others during their life.

Marcia's four choices for the young person mirror the choices that face the student study group facilitator regarding transparency about their teacher identity emergence with their peer study group program administrator. Among the four choices, three of them

result in the young person suppressing their emergent identity. Only one results in the young person selecting an identity not endorsed by authorities. We will return to this choice during the discussion section of this article when trying to understand the student choices revealed through the research study.

*2.8. Summary*

We cast a wide net to examine relevant research literature related to professional identity formation. Construction of identity is an activity closely interrelated between the individual and the social surroundings in which they interact. Since little has been published on this topic thus far with student study group leaders, this study's researchers were required to expand the search into various teaching-related positions. This literature will be instructive for this grounded research study of the outcomes that accrue to student study group facilitators.

**3. Methods**

*3.1. Sample*

Participants in the qualitative study were college/tertiary student Peer Assisted Learning (PAL) facilitators at an institution in the United States. These undergraduate college students were enrolled in a variety of academic majors. Their role was to serve as peer study group leaders in historically difficult courses (i.e., high rates of academic withdrawal and grades of D, F, and I). All 43 facilitators in the research study participated in a two-day (14 clock hours) initial training workshop focused on their job duties [7]. All of them (23 females and 20 males) were enrolled in the PAL facilitator professional development course, PsTL 3050 *Exploring Facilitated Peer Learning Groups* [8]. The academic profile of the PAL facilitators was an overall grade point average above 3.5 on a 4.0 scale. At a minimum, students were second-year (sophomores) or higher in their undergraduate studies. Data were collected from all 43 of them. The facilitators maintained a weekly journal of their PAL experiences to reflect upon their personal and professional growth. The facilitators were expressly requested to notice changes in themselves and record examples in their weekly journal entries.

*3.2. Procedure*

Data collection. The course instructor for PsTL 3050 was also the Primary Investigator (PI) for this qualitative study approved through the Institutional Review Board (IRB). Student names were replaced by pseudonyms to protect their privacy prior to data analysis. The PI was responsible for data collection. The data were obtained through the previously described required course (PsTL 3050). The PI created an online survey for students to complete. The survey questions were based on a review of previous data collection efforts by other peer assistance learning programs [3].

At the end of the professional development course (PsTL 3050), online surveys were administered to the PAL facilitators by the course instructor. The questions were selected based on previous data collection efforts. Students were informed that the course instructor would not know which responses were attached to individuals and would only know whether the facilitator had completed the survey. The survey must be submitted to earn a passing grade in the course. The surveys consisted of ten open-ended and two forced-choice items asking them to reflect upon their experience as a PAL facilitator, the possible impact they had on participating students in their sessions, and the program's impact on the facilitators' development. The survey data were coded for analysis in this study. The survey took between 30 to 45 min to complete.

Four additional data sources were consulted by the research study PI concerning these 43 students to triangulate data obtained with the online survey to validate consistency among the multiple data sources. First, the PsTL 3050 instructor, who was also the Primary Investigator (PI) for this research study, wrote observation notes on facilitators as they talked with each other during the PAL facilitator course and during individual office

meetings with them. Second, the PI read weekly private reflections required as part of the PAL course concerning students' experiences in the PAL position. Third, audio recordings were made of 25 PAL facilitators interviewed by the PI during which they responded to questions similar to ones in the online survey. Finally, reflections recorded in a book created by PAL facilitators were studied [32]. All four of these sources of information were reviewed by the PI for consistency with the data generated by the online survey. They were found to be consistent with one another. Due to the sheer volume of available information, the PI focused on the extensive online survey data for this research study, and other manuscripts are under development using the other four sources of data.

*3.3. Data Analysis*

This study employed a mixture of deductive and inductive analysis. It was deductive in that it was a replication of previous studies that investigated the professional identity emergence of study group leaders and tutors. On the other hand, our study was inductive since it identified responses in two other generic survey questions "*…describe and give examples of how you increased your personal skills, changed attitudes, or increased your confidence*" and "*…how did the experience of serving as a PAL facilitator impact you?*" Our study was also inductive because the identification of professional identity emerged from the data analysis through the previous survey questions. Since this was a grounded study, there were no research questions or hypotheses postulated.

Responses to the open-ended survey items were analyzed using a modified version of the steps outlined by Boyatzis [33] for inductive data analysis. Data analysis was conducted by the PI and a Graduate Research Assistant (GRA). Both of them had received training in qualitative and quantitative research methodology as part of their graduate programs. They approached the research analysis and interpretation as equals and resolved differences to each other's professional satisfaction. The total time spent on data analysis was approximately one month. In the first step, all responses were collected under each question. In the second step, the PI and the GRA independently identified themes across responses. They came together to discuss the themes and collaborate to reconcile any differences. A preliminary list of codes was co-developed. Codes were based on themes that emerged from the data. In the fourth step, the PI and the GRA independently double-coded the results, and each applied the codes to the data and then met to discuss their appraisals. They continued this process of double coding until they met an 80% or higher level of agreement to establish reliability (Boyatzis, 1998). After the collaborative process was completed, several initial minor themes were discarded due to having an insufficient number of PAL student facilitators associated with them. A final version of the major theme and several subthemes were established, and a final data analysis was conducted.

## 4. Results

Data analysis revealed a major theme of teacher identity emergence. This professional identity affiliation was how the students perceived themselves at a particular moment in time when serving as a PAL facilitator, but not a planned future career as a teacher. Approximately 60% of the facilitators reported a teacher identity for some or all of the academic term. At the beginning of the academic term, 20% of PAL facilitators were committed to a future teaching career, while the remaining 80% were intent on pursuing a future career as a researcher for a commercial firm or within postsecondary education.

The aforementioned PAL facilitators acknowledged they were breaking official boundaries to meet the needs of the students when they engaged in teaching activities and redefined themselves as teachers. Numerous times during the initial two-day PAL facilitator training workshop, the PAL facilitator course, and in individual meetings with the PAL program coordinator, the students were strongly warned not to engage in those behaviors but instead follow the formal job description established for the PAL facilitator position. These boundary crossings were carefully contemplated and nuanced when implemented. One PAL facilitator stated this succinctly:

"Although the PAL concept tends to look down upon "teaching" the students, I did come across times when the PAL concept just would not work, since simply none of the students had any idea what they were doing and had not been well informed of the material by their professors in order complete their assigned tasks. At this point, I would step in and use my knowledge of the course material in order to walk them through the problems, without giving them the answers, but safely guiding them on the right path".

As mentioned earlier, several other sets of information were used to triangulate the data revealed from an analysis of the survey. One was a set of reflections by PAL facilitators that were edited into a book [32]. Following are some passages by PAL facilitators on this issue of teaching during PAL sessions and dealing with the tension of violating major boundary prohibitions.

"The fact that we begin to care so much can sometimes put us in a tight spot. When students struggle, we want to alleviate that stress. To reassure them, we want to solve their problems—both figuratively and literally. The problem: that's not the role of PAL facilitator. A tutor, maybe. A facilitator, no. Have you ever heard the phrase that the right thing is often the hardest thing to do? It's kind of like that". (Walker, 2010, p. 1)

"What I like most about PAL is reaching out and teaching students—for the sake that I want to be a teacher—and knowing that something I did helped them learn". (Walker, 2010, p. 2)

"Even though that's not what we're supposed to do [providing brief lectures before exams]—we should be redirecting questions to the students' peers—it's hard. It's hard to know where you can cross the line and where you can't in regard to how we're supposed to help as a PAL facilitator". (Walker, 2010, p. 2)

"I know we're not supposed to re-teach or do that in PAL, but I didn't know any other way". (Walker, 2010, p. 8)

"I have definitely crossed the line between being a facilitator and being a teacher, but it's like, what can you do?". (Walker, 2010, p. 12)

Within this major theme of manifesting a teacher identity full- or part-time during the academic term when serving as a PAL facilitator, several subthemes emerged for smaller percentages of the facilitators: (a) requested additional clarification of facilitator role, (b) provided limited instruction, and (c) assessment of participant needs sometimes led to other discussions. The theme and subthemes were based on self-reports by the facilitators to generic questions from the survey instrument.

*4.1. Subtheme A: Requested Additional Role Clarification of the Facilitator Role*

Nearly a quarter of the facilitators requested clarification of their PAL role, especially related to the level of control and authority they should exhibit during the study group sessions. The responses indicated that some facilitators found the official PAL approach confusing and unwieldy. Several facilitators described the need for more guidance on the appropriateness of teaching when needed. These comments indicate that revision of the PAL facilitator initial training workshop and the subsequent PAL facilitator course curriculum need to be revised accordingly. The sophisticated and nuanced experience of serving as a PAL facilitator was not addressed during these training activities by the professional staff. It is also another reminder of why it is helpful, if not essential, for the PAL program administrator to serve as a facilitator on an occasional basis.

"It would have been beneficial if we would have been explained in detail about how much are we supposed to teach in class. I know we are not hired as Teaching Assistant only, but there were times in the beginning where students were confused about something that I knew but I restrained from explaining that in detail".

"I really think there needs to be some time spent explaining how we are supposed to "fade away" as facilitators. What I mean by that is that when this [PAL training] started we were told that by the end of the year someone would be able to walk into the classroom and not know the facilitator actually was, well, I think everyone could tell I was the facilitator. . . .So what is this anyway?...I feel a contradiction in the program and I think we need to be taught better or know better which one we are supposed to do, and how to do it".

"I wish that we could have addressed the issue of homework. I know the policy towards it is very straightforward, but I still would like a discussion about it and what role it should play in PAL, if any. I don't believe the policy should be so black and white as it is".

Some respondents expressed the need for direction on specific behaviors, such as "when to ask questions to the students and when to not directly answer what they want to know", whereas others described more general confusion about the level of control they should exert within the sessions. "I think there may have been a bit more emphasis placed upon ways to structure a session without taking too much control". This uncertainty led facilitators to make their own judgments within sessions while still trying to keep to the tenets of the PAL program.

### 4.2. Subtheme B: Provided Limited Instruction to PAL Participants

About a quarter of the facilitators adapted their sessions to provide short lectures when they perceived a need. This directly opposed a specific rule from the initial training workshop to never engage in lecturing since PAL sessions were about reviewing content and not presenting it. Three facilitators described the impracticality of never teaching students directly.

"I generally tried to help the students to learn how to help each other when possible, but in a College of Liberal Arts course, if you don't know the material, you don't know the material. In cases where nobody knew much of anything I adapted my role into a more traditional teaching role for the sections that they were struggling with".

"It was hard for me to gauge when it would be OK for me to tell a student because I've had instances this semester when no matter how hard the student tries, . . .no one has any idea how to answer the question. I guess I would have liked to see more role-playing or instruction about how to deal with these situations".

"Initially, I found it rather difficult to not "teach" at all. Lets face it, the vast majority of the questions that I handed out, I could do within a few minutes. Although, I quickly realized how much the students actually remember and understand the material more when not just told the answers. Of course, you get the typical "Why don't you just tell me the answer?" or "Come on, you know the answer so I can just hear it now, and be able to do it on my homework." But after facilitating, for example, molding the question into an easier example and following with, "How did you figure that question out" such that they could apply the same concept with the new question, you can just tell how much the material starts to "click" to them".

One respondent expressed how, although s/he had to adapt her sessions from the initial training s/he received, s/he believed this flexibility was in the best interest of the students.

"My students didn't usually go to their TA session, so throughout the semester I had to become a little more of an explainer/clarifier then I would've liked, but I think it truly was what was best for the students in my sessions, so I don't think I would've done it differently".

Another respondent reported that as students gained more experience and confidence, the facilitator did less instruction. This highlighted how participants' responses and engagement level in the study sessions shaped the facilitator role.

> "I adapted my role throughout the semester to give them extra guidance when they needed it and to hold off when one of their classmates could explain the material better. As they became more comfortable in the class, I was able to do less as I could ask other students to approach the board to explain a subject".

The facilitator's approach to teaching evolved over the course of the semester as they tried different methods and learned what worked best for their students and reduced direct instruction as a result. One-tenth of the facilitators explained their choice of occasional direct instruction as a result of the PAL participants' expectations of them. The inclination to teach could also be reinforced by students' expectations about their roles. Several respondents cited the pressure students placed upon them to provide answers directly and their attempts to get around these requests. "The students who asked me for help and answers all the time, I referred them to their group members. . .and then the group asks me the answer, I would tell them, and they would either be happy, or try again".

### 4.3. Subtheme C: Assessment of Participant Needs Sometimes Led to Other Discussions

Finally, although the didactic and subject matter expert orientations seemed to be the most adopted approaches, about 20% of facilitators described instances where they focused on the facilitator's engagement with the students, including understanding their worldviews and supporting them in personal problems.

> "Sometimes my role as a facilitator changed from that of someone helping with material in classes and the students just wanted help with navigating their paths through the university. As an upperclassman I know I was able to give help and advice to the students as someone who had already experienced the struggles of registering for classes, choosing professors, and planning out the next four years. It was sometimes appropriate to take on the role of a helpful friend who could give advice and step away from the material we were focusing on".

As previously mentioned, another book was composed of PAL facilitator reflections about their role. Some of those entries provide triangulated confirmation of this subtheme.

> "Students and faculty may have a hard time defining your role, and as a result, facilitators wear many different hats. Advisor, friend, confidant, reference—you name it. With the goals of PAL and your own instincts as guidance, you can confidently choose the hats you wear". (Walker, 2010, p. 15)

In this role, the facilitators approached the students as more of a peer than as a facilitator or instructor. Ultimately, in deciding what level of teaching and facilitating to do, many facilitators kept in mind the guidance from the PAL program while also relying on their own judgment in the sessions. These data, thus, suggest that the facilitators thought critically about their professional identity and understood the need for flexibility in balancing the roles of didactic expert, subject matter expert, and pedagogical expert.

### 4.4. Summary

The emergence of the professional teacher identity was the dominant theme of this grounded research project. Within this larger theme, several permutations of the theme were expressed: request for additional role clarification of how to do their jobs, how to provide limited instruction through mini-teaching to the students, and finally, assessing student needs of the PAL participants sometimes led the facilitators into other conversations. The PAL facilitators felt compelled to cross explicit role boundaries to meet the immediate needs of their session participants.

## 5. Discussion

### 5.1. Emergence of Teacher Professional Identity

The emergence of a teacher professional identity was a surprise to the researchers since this was a grounded research study based on facilitator responses to open-ended general questions about their experience in the PAL job role. Only 4 of nearly 2000 publications about peer learning groups [3] identified teacher identity emergence [9–12]. For years, it was common for the PAL PsTL 3050 course instructor to hear facilitators refer to themselves as teachers as they worked with their students. It was erroneously assumed that this was not indicative of professional teacher identity and role in violation of the PAL-prescribed boundaries. In this study, nearly 60% of the facilitators emerged with a professional identity as a teacher for all or part of the academic term. This identity emergence occurred along a time continuum throughout the academic term. It was not an outcome after the conclusion of their PAL experience.

Identity is socially constructed with others (PAL participating students and other PAL facilitators) and not theoretically selected by the facilitator in isolation. The informal community of practice with other facilitators outside the view of PAL administrators, weekly reflective journal entries, and the final reflective writing activity at the end of the academic term were all components of the ecosystem that fostered identity emergence. While some already planned on a future teaching career, many others felt "pulled" into this identity. PAL facilitators rewrote and expanded their job descriptions to accommodate their new teacher identity in response to their assessment and understanding of their students. This is like the experience of the PLTL student leaders reported by Clark and Raker [11], who felt the need to redefine themselves in response to the learning environment created by the explicit needs of the study group participants.

As discussed earlier in this article, Marcia [31] explained that identity often emerges from a crisis and the commitment to resolve it. People in such a crisis have four choices to make: (a) foreclosure, choosing to follow directives of authorities rather than meet the needs of others; (b) moratorium, authorities are questioned regarding role boundaries but no action is taken other than to follow directives of the authorities; (c) identity development, the person chooses a new identity which is risky since it is not authorized by the authorities; and (d) diffusion, the person lacks commitment to most things in life and blindly follows authorities and others in their life. Considering the choices made by many PAL facilitators, they appeared to follow option "c" with the development of a new identity that was perceived as risky since they were clearly acting against the boundaries established by the PAL administrators.

A previous model for understanding the emergence of teacher identity among novice and experienced teachers sheds light on the emergence of this new identity for the PAL facilitators. The PAL facilitators mirrored the behaviors of the novice and experienced teachers. Beijaard et al. ([28,29]) found that most teachers surveyed favored didactic and subject matter expert orientations. PAL facilitators generally followed didactical approaches but fell into the dominant role of subject matter expert early in the academic term. After the initial period in the academic term, most moved toward the bottom of the diagram by shifting from a subject matter expert to a humanistic orientation of relationship building and facilitating sessions by transferring authority and cultivating ownership of the sessions to the participants. This was like Beijaard et al.'s [28,29] findings that novice teachers were located at the top of the model. They clung to their subject matter expert role, while experienced teachers exhibited more behaviors associated with humanistic and facilitative expertise. They empowered students to co-create their learning environment in which they developed their own subject matter expertise. The line formed between subject matter and the didactic/facilitative expert creates a continuum for the transfer of authority and ownership of the learning environment to the students.

Beijaard et al. [28,29] found that nearly half of the teachers in their study endorsed more than one of the three categories of teacher identity [Figure 2]. Accordingly, they argued that these categories should not be seen as dichotomous but rather fluid depend-

ing on the teaching context, relevant learning experiences, and the teacher's biography (Beijaard et al. [28,29]). This study reinforces fluidity, as facilitators integrated the PAL training model (relevant learning experiences) with the needs and expectations of the students and their perceptions of their role (classroom context and teacher biography).

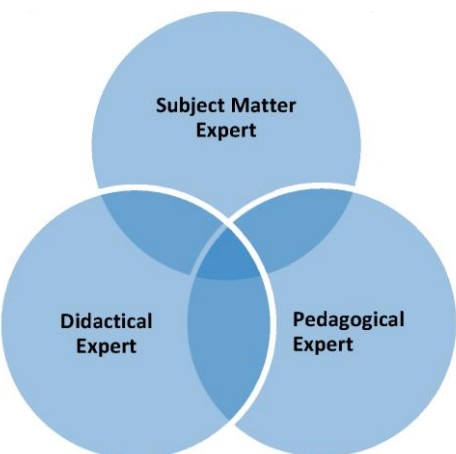

**Figure 2.** Model of Behavior by Novice and Experienced PAL Facilitators.

*5.2. Requested Additional Clarification of Facilitator Role*

Facilitators in this study were guided by a formal PAL training model that emphasized the facilitation of student study groups. They received explicit instructions not to lecture under any circumstance and to follow their official job description, which focuses on facilitating academic discussions among the participating students. Little was provided during training on how to establish relationships with study session participants to assess their learning needs and adapt PAL sessions accordingly. Facilitators sought to implement the PAL principle of transferring authority and ownership of the PAL sessions to participants but found that initial training failed to provide practical strategies to do so. Based on the data, it appears that facilitators more likely engaged in short bursts of clarifying or explaining rather than delivering longer lectures.

*5.3. Provided Limited Instruction to PAL Participants in Their Best Interest*

Once they began their work as PAL facilitators, they found the PAL role boundaries often inhibited addressing their students' needs. Early in the academic term, some facilitators assumed the role of subject matter expert and engaged in direct instruction when the students were unable to understand the content or solve the problems. Most of them reported that the incidence of direct instruction lessened or disappeared as they gained experience with the facilitator role during the academic term. Based on the data, it appears these incidents of direct instruction were short bursts that occurred within the PAL sessions early in the academic term. A more accurate term to describe these events was as an occasional clarifier/explainer rather than a lecturer. Sometimes, the PAL facilitators expressed regret for breaking the boundary rule of no lecturing, but based on their professional judgment, they did so since they perceived study session attendees needed the help. As they matured in their role as PAL facilitators during the academic term, they moved from being subject experts to, instead, focusing more time on employing.

Several other factors may have played a role in this boundary-breaking: (a) facilitators knew more of the needs of the participants since they were of similar age and life experiences; (b) facilitators were concurrently students in other classes and knew what they personally needed to learn and academically succeed; and (c) they were effective in assessing the needs of the students through analysis of their body language and individual conversations with them. Dunn et al. [21] found that college teaching assistants naturally replicated the teaching practices they experienced previously and concurrently as students. Due to their inexperience in the role and inadequate initial training, it was easy to default

to traditional instructor behaviors. The same was true for the PAL facilitators. They found it easy to default back to their experiences with lecture-based classrooms in secondary and postsecondary courses. This issue of boundary crossing of lecturing was aptly described in the article by Brown et al. [10] titled, "We were told we're not teachers...it gets difficult to draw the line: Negotiating roles in peer-assisted study sessions (PASS)". Dr. Kim Wilcox was a former national training director for Supplemental Instruction-PASS. During SI-PASS program administrator training workshops, he commented how easy it was for all the SI-PASS training protocols to vanish when the SI-PASS leader closed the door to the classroom for early SI review sessions since it was just them and the anxious participants. They too often defaulted to an easier and more familiar question-and-answer format (Personal Communication, 1998).

*5.4. Teacher Professional Identity Influenced by the PAL Work Environment*

The teacher professional identity was fostered by the facilitator's choices and experiences in the PAL work environment. The following sections of the article explore more deeply the learning environment, which challenged both the facilitators and participants. It is a rich and complicated ecosystem.

### 5.4.1. The PAL Sessions Setting Physically Resembled a Regular Class

Due to the mandatory attendance of PAL sessions, it was common for them to have 20 to 30 regularly attending participants. This was a common size for other college courses for the facilitator and the participants. Participants came into a PAL session without deep knowledge of the PAL philosophy but to have their academic needs met. The meeting space for the sessions was most often the same classrooms they attended for their regular class lectures by the instructional staff. This perhaps contributed to confusion for the PAL participants since it was the same physical location but with a different purpose.

### 5.4.2. PAL Participant Expectations

The PAL participants played a role in pulling the facilitators to exhibit teacher behaviors. At the beginning of the term, some students requested information from the facilitators and expressed frustration when it was not provided. Their implicit requirement to be active learners in the review sessions was unanticipated by most of them since they were not aware of the nuanced PAL philosophy and job expectations placed upon the facilitators.

### 5.4.3. Community of Practice (CoP)

Outside of meetings conducted by the PAL administrator, it appeared that facilitators created their own community of practice (CoP) to provide peer conversation and support for their job experiences outside of meetings conducted by the PAL administrator. The researchers of this study postulate that this hierarchy-free environment created a nurturing environment to explore identity and share strategies for extending their job descriptions to better meet student needs. CoPs can exist within formal institutions but are invisible to all except members of the group [14,15]. CoPs were cited by Dam and Blom [22] as the location where secondary school student teacher's teacher identity emerged in conversation with other student teachers. James and Diment [18] found that teacher identity for British tutors emerged from informal conversations with other tutors. The tutors went underground into a space neither validated nor recognized for these conversations since "...the practice in question may be somewhat at odds with orthodox or officially sanctioned reifications..." (p. 420). As identified by a study of novice high school teachers, a person's identity is socially constructed with others and not theoretically selected in isolation [27].

### 5.4.4. Frequent Reflections of Facilitator Experience

The PAL program required facilitators to complete weekly journal reflections about the growth of the participants and their personal and professional growth. In addition, reflections about PAL activities were discussed weekly during the PAL course, followed

by an extensive written reflection about the PAL facilitator experience at the end of the term. These reflection activities were not purposed for identity emergence but rather as a common practice of having student paraprofessionals consider their experiences more deeply. Several other studies identified similar reflective activities linked with teacher identity emergence [22–24,26].

### 5.4.5. Facilitator Assessment of Participant Needs for Other Conversations

As facilitators developed relationships with students and assessed their personal and academic needs, new conversations emerged in the sessions. These could be interlaced with the academic review. Issues such as navigating the complicated campus environment, understanding critical issues for completing graduation requirements, and referrals to campus student resources such as advising, counseling, and others were as important for academic success as preparing for the next exam.

### 5.5. Model for Professional Identity Emergence of PAL Study Group Facilitators

Compared with the previous professional literature reviewed, our PAL facilitator behavior was most consistent with the model by Beijaard et al. [28,29], which examined novice and experienced secondary school teachers [Figure 1]. It was seductive for PAL facilitators to exhibit subject mastery and engage in short or long bursts of teaching behavior. This was similar to the experience of novice secondary teachers. Despite their recent PAL training workshop experiences, it was easier to exhibit the behaviors of their classroom instructors initially. These behaviors were the most familiar to the PAL session students and were often preferred, with PAL sessions taking the form of question-and-answer interchanges rather than the desired learning environment of participating students taking on more responsibility for their learning and developing their metacognitive skills so that they could become autonomous learners for this and other courses during their academic career. As noted earlier, once the PAL session room door closed, it was a challenge for many PAL facilitators to exhibit the behaviors that they had recently been trained in when faced with a room full of anxious students who were worried that their academic careers and future employment prospects were at risk if they failed the course.

Consistent with the behaviors of experienced secondary school teachers who move to a role of facilitated learning and away from didactic lecture-based pedagogy that displays subject-matter expertise to the students [28,29], most PAL facilitators traveled in a similar direction over the short time of a single academic term to behave more like the experienced teachers in the Beijaard study. A question arises about how the PAL program administrators can help the facilitators make that transition more quickly and with confidence that they are meeting the needs of all stakeholders, including the classroom instructor, PAL program administrators, PAL participating students, and the integrity of the PAL group program. That will be explored later in this article.

The following professional identity model [Figure 3] is influenced by pioneering work by Alexander Astin and his colleagues explaining the changes that occur during the college experience for students [34]. In its basic format, this model is composed of three components located along a continuum: input variables that the student brings to college, environmental variables the students experience while at college, and the outcomes from the interaction of their input variables with the college experience. This model [Figure 3] is also influenced by the work of other scholars who conducted research studies on student peer study group leaders [3], previous studies that led to the postulation of a model to explain changes in future vocational choice [3], and a model to explain the emergence of leader identity while they were serving as PAL facilitators.

This model seeks to identify the complex experiences of PAL facilitators, what they bring to their experience as a facilitator, and potential outcomes from that experience that have been identified by other researchers [3]. The following are the three clusters of variables that have an impact on students: input, college environment, and outcomes. The college environment is broken into two categories: bridge involvement by the student

upon immediate arrival at college and intermediate involvement for the rest of their college experience. Following is a graphical model to identify the complex ecosystem in which the PAL facilitators operate during their college experience as a student and as study group leaders [Figure 3].

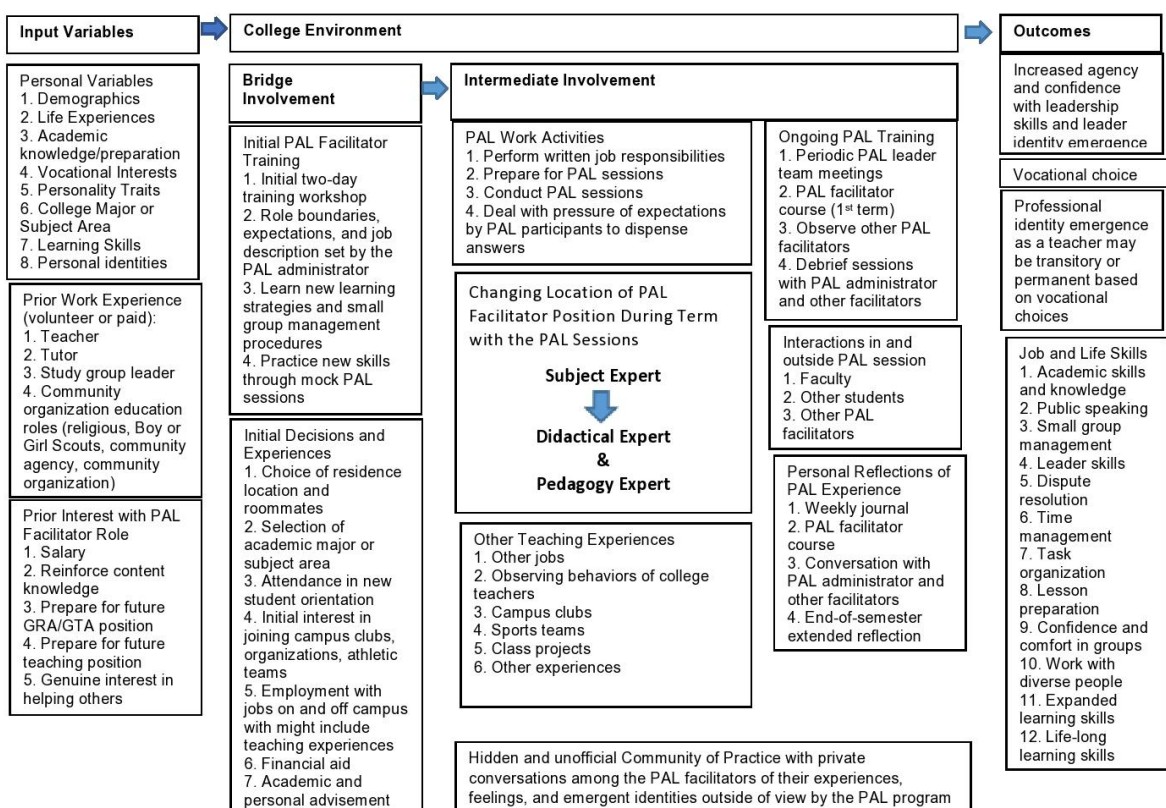

**Figure 3.** Professional identity development model for peer study group facilitators.

### 5.5.1. Input Variables

Input variables are what the students bring with them to their college experience. They include their personal demographics, life experiences to that point, and personal identities, which could include an inclination to a teacher professional identity based on prior experiences. This leads to the second cluster of inputs, which are composed of voluntary and paid work experiences. For some students, they have prior experience with teaching. The final cluster of inputs is their prior interest in seeking the position of a PAL facilitator.

### 5.5.2. College Environment Variables

When Astin [34] analyzed the college experience, they identified more than 100 variables. With this model [Figure 3], we have focused on those that are most relevant for the PAL facilitators. These variables are broken down into two sequential groups: bridge involvement and intermediate involvement. Bridge involvement variables are those that the student either chooses or experiences after their arrival at college. The initial decisions and experiences cluster are those of a more personal nature regarding the choice of living location, the initial meeting with an academic advisor, attendance at new student orientation, and more. The second cluster of bridge involvement variables is related to the initial PAL facilitator training activities before they begin working with students in the target course.

The second set of variables are those related to their intermediate involvement within the college environment, primarily directly related to their involvement as a PAL facilitator. As mentioned above, Astin [34] and others identified more than 100 involvement variables that have an impact on a student. Within these intermediate involvement variables

are seven clusters: PAL work activities, location within the Beijaard et al. [28,29] model [Figure 1] for behavior within a PAL session as discussed earlier in this article, other teaching experience, ongoing PAL training activities, interactions in and outside PAL sessions, personal reflections of the PAL experience by the facilitators, and involvement with a hidden and unofficial community of practice with other facilitators.

The final set of variables are those that accrue because of the PAL facilitator experience at the conclusion of college. Some of these are vocational choice, increased agency, professional identity, and a diverse set of new and improved job and life skills. As mentioned above, more than 100 research studies have identified this broad range of skills by student peer study group leaders. Separate previously published studies by the authors of this article identified vocational choice and work skill acquisition [3].

For purposes of this article, a detailed exposition of all the variables related to the PAL facilitator experience is not provided. As stated above, some articles have already been published [3], with others under development.

## 6. Implications for Revising the PAL Model

While maintaining the primary focus of meeting the academic needs of struggling students in tough classes, expand the opportunities for the program to be a more intentional incubator of development for the facilitators. Many studies have identified the benefits of student-led study programs for participants and student leaders [3]. Imagine the results if the study program approach was more intentional for the facilitator's growth.

### 6.1. Discuss Identity Formation during the Initial PAL Training

First, discuss the PAL facilitator's professional identity development during the initial PAL training workshop. During this discussion, share the Beijaard et al. [28,29] model [Figure 1] and the three categories of behaviors they need to exhibit. This open and healthy dialogue can avoid inadvertently causing PAL leaders to converse covertly with one another when they operate outside the PAL model boundaries. Introduce the vocabulary for identity professional development and allow the students to think about it when they reflect on their job experience. They would view the job description as a guide for setting expectations rather than a precise list of approved behaviors.

### 6.2. Reorganize the PAL Facilitator Training Curriculum

Second, building upon this open dialogue of professional identity development, the initial training workshop and the training materials could be reorganized around the three areas outlined in the Beijaard et al. model [28,29]. These would not be equal in length but would provide a visual model for the PAL leaders to use in their sessions. One component is displaying subject matter competence and how to develop it in the participating students in the study groups. The second component is a humanistic/pedagogical approach to learning. This would contain strategies for assessing student needs, developing empathy with students, and flexible lesson planning and responses during the PAL sessions based on responses by the participating students. The final component to display during PAL sessions is didactic/facilitated learning activities. This includes using active learning activities, models for peer cooperative learning groups, and other methods to shift responsibility for learning to the students rather than relying upon the PAL facilitator. Past experience by the PI who taught the PAL facilitator course encourages sharing explicit learning theory in the initial training of the facilitators. The PAL facilitator course required weekly readings and subsequent discussions on assigned research and theory articles relevant to their role as the study group facilitator.

### 6.3. Endorse the Clarifier/Explainer Role

Third, the PAL program administrator endorses the part-time role of clarifier/explainer for limited use with the study group students when necessary. In this research study, students shared that these teaching times were of short duration except for the beginning of

the academic term when they depended on this activity due to their perception that the students were not yet engaging fully in the learning activities during the study sessions. To help deal with the political consequences of referring to this activity as teaching, call these instances clarifier/explainer. New training strategies would be needed to model its use with simulated sessions during PAL training workshops or during team meetings throughout the academic term. It is recognized that this recommendation opposes boundaries established by most of the major national or international peer learning programs such as Supplemental Instruction-PASS. It is best if the study group participants perform the role of clarifier/explainer working in small groups. However, this role may be necessary at the beginning of the academic term by the study group facilitator.

### 6.4. Expand Knowledge of Student Development Theories

Fourth, it would be useful for the study group program administrator to expand their knowledge of student development principles and theories. If not a member, explore professional associations such as the American College Personnel Association, Council for the Advancement of Standards in Higher Education, National Resource Center for the First-Year Experience, National Association of Student Affairs Professionals, and First-year Students and Students in Transition. Membership offers professional journals and access to member resources, including discounted registration fees for regional and national conferences. The study group program administrator could also contact campus professionals involved with advising, counseling, and others for possible involvement in PAL facilitator training sessions and for their own professional development.

### 6.5. Increase Guided Reflection Activities by the PAL Facilitators

Fifth, increase the use of intentional and guided reflections to increase the opportunity for the facilitators to explore identity development (vocational choice, leader perception, and professional identity) and identify skills they are developing. Many campus programs already include reflection activities. Consider increasing their frequency to weekly. Make specific writing prompts on different weeks, such as the three identities mentioned above and the skills they are gaining. An extensive final writing reflection at the end of the term would be a capstone experience for facilitators to be introspective about their personal learning discoveries. Research has shown that intentional reflection is a major catalyst for identity development. If the amount of personal and professional development occurs for PAL facilitators as demonstrated by this study, imagine the outcomes if the reflection process was more intentional and extensively discussed as a group with the other facilitators and the PAL administrator [35]. Building upon written reflective journal entries could be the construction of electronic portfolios that could more creatively reflect the lessons learned through the PAL experience that could be displayed for potential employers. Portfolios integrate features of reflective writing, resumes, and other evidence of personal and professional growth.

### 6.6. Promote Opportunities for PAL Facilitator Personal and Professional Growth

Sixth, explicitly promoting personal and professional growth opportunities for facilitators during the hiring process may increase the number of facilitator applicants. Establishing in students' minds that involvement with the study program is a co-curricular experience for participants and facilitators helps shed notions of remediation and redirects to the rich outcomes available. Include in reports to campus administrators results concerning facilitator development and grade and persistence improvement for the participants. Use surveys, focus groups, individual interviews, and other means to collect data and stories from the facilitators. Rather than positioning the study group program as a stand-alone program, reframing it as an incubator for student learning and student development outcomes positions the unit at the institution's heart and can increase financial stability and support during difficult financial decision-making by upper-level administrators.

### 6.7. Provide Continuous Feedback to the PAL Facilitators

Seventh, provide continuous feedback to the PAL facilitators regarding their session planning and execution of those plans during PAL sessions. This feedback can occur through direct observation of PAL facilitators during their sessions by the PAL administrators and other PAL facilitators. Following these observations, confidential conversations could occur in which both the observer and the facilitator have a candid conversation regarding what happened during the session and how it followed or deviated from the initial PAL session plan. These observations should occur throughout the academic term. We suggest at least seven times each term. The authors of this article have received reports from program administrators of programs like PAL that they only observe one or two sessions each term due to increased administrative responsibilities, supervision of additional learning assistance programs, and other duties. As noted earlier in these recommendations, having all facilitators complete weekly journal entries on their experiences with feedback provided by the PAL administrator is invaluable for affirming behaviors by the PAL facilitators that resonate with the PAL model and noting how to deal with choices during the sessions. Providing affirmations to these young facilitators is essential for their personal growth in what can be a sometimes tension-filled learning group session.

### 6.8. Encourage a Relationship with the Teacher Preparation Program

Finally, encourage the campus teacher preparation programs to actively recruit PAL facilitators for their programs. Other research studies have indicated that the PAL-like experience encourages the facilitators to consider future careers as teachers [3]. Reversely, teacher education programs could encourage prospective teacher candidates to apply for positions as PAL facilitators. A symbiotic relationship could be formed between the campus learning assistance programs and teacher education preparation.

## 7. Limitations

This qualitative research study has several limitations. First, this study focused on PAL facilitators' perceptions of their PAL job experiences. It is possible they made errors in the interpretation of the PAL experience. The responses by PAL facilitators were subject to perceptual recall, which reflected their interpretations, judgments, and potential bias. By its nature, this research is subject to the limitations of the self-reported data of the survey. While we triangulated the survey findings with separate reviews of their weekly journal entries, individual meeting notes, and recorded interviews for a PAL podcast, we did not conduct additional follow-up interviews or record PAL sessions for further analysis. Second, most of the college courses served by the PAL program were in science and mathematics. It is possible that a wider range of academic subjects could have fostered different results. Third, all PAL facilitators were undergraduates, as were the participants, and the selected courses for PAL were at the lower division of the undergraduate curriculum. It is possible that a different experience would have resulted from having graduate students serve as the PAL facilitators or if the classes served the upper-division undergraduate or graduate level. Finally, PAL sessions at an open-admission institution might have derived different interactions than those in this study.

## 8. Future Research Areas

Whereas Beijaard et al. [28,29] identified the effects of the teaching context in fostering teacher identity, they did not fully emphasize the role of student expectations. Our study indicated that student expectations and needs strongly influenced the PAL facilitator's approach. Accordingly, future research might also look at the role of student expectations on facilitator identity development and how they intertwine. This would create a more dynamic model examining their interaction.

Another study could investigate the life experiences and attitudes that encourage PAL facilitators to exhibit professional identity development more often associated with experienced teachers (Beijaard et al. [28,29]). Usually, maturation leads novice teachers to more

complex models of identity. Could the relative closeness in age between the PAL facilitators and the PAL participants allow them to operate by the model of Beijaard et al. [28,29]? Understanding the development of professional identity for these paraprofessional student employees could be an important contribution to the professional literature on teacher professional identity development.

A challenging area for deeper understanding is the inner workings of communities of practice (CoP) formed by student-led study group facilitators. By its very nature, it is difficult to study an environment that is privately held by the group members and operates outside the observation of the professional staff or faculty members. Previously cited research studies affirm the powerful influence of CoPs upon student tutors and small-group study group facilitators. However, little is known about how they operate and the private conversations that occur among the students. Rather than operating as a separate, and sometimes opposing, influence upon the student paraprofessionals, how could they be supported by the professional staff? Understanding how these CoPs operate within the learning assistance context could be valuable for supporting CoPs that operate in other areas within academic and student affairs, such as peer mentors, student orientation leaders, residence-hall assistants, and Graduate Teaching or Research Assistants.

## 9. Conclusions

This study focused on student paraprofessionals who facilitated study groups for academically challenging college courses. A grounded qualitative research study of these student facilitators at an institution identified teacher professional identity development by them in ways not expected and against written policies for their behaviors. This largely hidden world of professional identity emergence was surprising since it broke the boundaries of behavior established by the study group program administrator. Rather than suppressing this identity formation by the student leaders, recommendations were made so they could openly talk about this identity with the PAL administrator and express it in ways that would be acceptable to the instructional staff. This study is a reminder of the richness of the student study leader experience for their personal and professional development. Development of trusting relationships among the program staff and the student paraprofessionals could open up other dimensions of the often unseen world of identity emergence and the development of even more helpful strategies to help students in the study groups earn higher grades, persist until graduation, and achieve outcomes for personal and professional development by the participants and the student study group facilitators.

**Author Contributions:** Conceptualization, D.R.A.; methodology, A.R.H.; validation, D.R.A. and A.R.H.; formal analysis, D.R.A. and A.R.H.; investigation, D.R.A. and A.R.H.; resources, D.R.A.; data curation, D.R.A.; writing—original draft preparation, D.R.A. and A.R.H.; writing—review and editing, D.R.A.; visualization, D.R.A.; supervision, D.R.A.; project administration, D.R.A. All authors have read and agreed to the published version of the manuscript.

**Funding:** This research received no external funding.

**Institutional Review Board Statement:** This study was exempt from the Institutional Review Board. IRB# 0801E24406 issued 15 January 2008.

**Informed Consent Statement:** Written informed consent has been obtained from the students) to publish this paper.

**Data Availability Statement:** Data for this study are available upon request from the corresponding author.

**Conflicts of Interest:** The authors declare no conflict of interest.

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
