# Peer review of "Professional Teacher Identity on the Boundary: Student Study Group Facilitators Negotiating Their Identity"

_education, doi:10.3390/educsci13121210_

Round 1

Reviewer 1 Report

Comments and Suggestions for Authors

Education Sciences Manuscript: education-2608045

Dear Author,

Thank you for submitting your work to Education Sciences. Unfortunately, in its present state, the paper is not suitable for publication. Below are a few points that are hoped to help you to improve your paper prior to further processing.

Wishing you success.

Abstract

Better articulation of the abstract is required.

Too many keywords are used.

Introduction.

·       There is no need to repeat the abstract word by word in the introduction.

·       On line 39, the term ‘rigorous’ may not be ideal. Other courses in the liberal arts for example may be rigorous but not in demand. The author ought to clarify the use of the term rigorous throughout the paper. Do you mean intensive rather than rigorous?

·       The aims of the paper are not clearly articulated. The paper appears to be a recount of an experiment held within the selected institution. A research question driving the study also appears to be missing.

Literature Review

·       The literature review is very descriptive in nature. It lacks any type of critical engagement with the literature which gives the piece a narrative style, almost a summary.

·       The key terms are never clearly defined. Due to the nature of the prosaic review it risks having the reader wonder about the relevance of the different sections. The notion of communities of practice (CoP) is interesting. How does this influence identity as an individual moves from being a novice to become an expert? How is this notion relevant to the PAL programme? Why is expertise necessary especially in the light that no teaching should be done. The paper should address these questions and other related to the different sections to improve the quality of the paper.

·       The author should clarify if they are talking about the identity of teachers or students, or facilitators throughout. In section 2.7 the identity of students is mentioned. How is this relevant? Is the author talking about those students who act as facilitators?

·       This section ought to be revised in its entirety to make sure that the aims of the paper (that are poorly defined) drive the literature review.

·       A conceptual model driving the study is missing. Hypothesis formulation is missing,

Methods.

·       This section needs to be re-written so it is more coherent.

·       The research design should be properly outlined. From what comes across in this section, the study is flawed a priori. There does not seem to be a control group and the data appears to have been collected at the end of the course. A pre/post-test design seems to be missing.

·       With reference to the questions asked, where the scales used previously validated?  Are they standardised scales or are they simply end of course questions turned into a measuring scale? The answer to these questions should naturally be no because the questions used (based on the analysis) are not of a quantitative nature. The author ought to clarify what type of data was collected and how. A list of the questions or at least a sample should be presented.

·       The author refers to other four data sources (line 392). It is not clear what these sources are.

·       The author also failed to note which variables are observed and what the expected relationships are.

·       Sampling. What type of sampling was used?

·       What were the inclusion criteria for the sample selection? Where any inducements offered to participants?

Results

·       The data analysis method needs to be clarified. Since the term survey was used, one would expect quantitative data, however, the author also refers to open ended instances. It is therefore unclear what methods were used to assess any quantitative aspects.

·       How were the additional resources triangulated?

·       The main theme out to be articulated better.

·       Are references to Walker (2010) emerging from the excerpts or are the related to other sources?

·       The excerpts used miss the pseudonym and are not presented correctly. The author is encouraged to improve this.

Additional comment:

Is Figure 1. Model of professional identity among novice and experienced teachers in any way used as a conceptual model to drive the study? If yes, how is it aligned with the research aims and the (missing) research question? Perhaps the author could revise this since this model also features in some way in the results, although Figure 2 refers to PAL facilitators and not teachers.

The outcomes featured in Figure 3 are useful and interesting. They merit a better discussion particularly linked to how the results from this study may contribute to the literature.

Perhaps proof reading the document by a native speaker prior to submission would be useful.

Comments on the Quality of English Language

Perhaps proof reading the document by a native speaker prior to submission would be useful.

Author Response

Our response to reviewer #1 is attached. We are grateful for the time and effort with your careful review and the recommendations for improvement. Thank you.

Reviewer 2 Report

Comments and Suggestions for Authors

The paper explores an aspect that is neglected, namely the professional development and professional identities of peer facilitators. Thus, it contributes immensely to the understanding of the forces that shape the identities and role behaviours of peer facilitators.

Author Response

Our response to reviewer #2 is attached. We are grateful for the time and effort with your careful review and the recommendations for improvement. Thank you.

Reviewer 3 Report

Comments and Suggestions for Authors

Thank you for your paper.

The paper is coherently structured, well written, and focused on furthering the field of interest. The topic seems relevant and well embedded in existing research. Well done.

Some minor corrections or additions would be needed:

You might include the college’s country for context in section 1.1.

Typo in figure 1 and figure 2: „Subject Matter“

Figure 2 is not well aligned

Figure 3 is low quality and not well aligned. The word „Matter“ is dropped from the figure („Subject Matter Expert à Subject Expert“)

Section 4: Some paragraphs seem to be quotes from the data, although formated as regular text.: „Although the PAL concept tends to look…“ etc.

Comments on the Quality of English Language

Please review spelling in figures.

Author Response

Our response to reviewer #3 is attached. We are grateful for the time and effort with your careful review and the recommendations for improvement. Thank you.
